# Integrated Numerical-Experimental Assessment of the Effect of the AZ31B Anisotropic Behaviour in Extended-Surface Treatments by Laser Shock Processing

**Ignacio Angulo \*, Francisco Cordovilla, Ángel García-Beltrán[ID], Juan A. Porro, Marcos Díaz[ID] and José L. Ocaña[ID]**

UPM Laser Centre. ETS Ingenieros Industriales. Universidad Politécnica de Madrid. C/José Gutiérrez Abascal, 2, 28006 Madrid, Spain; francisco.cordovilla.baro@upm.es (F.C.); agarcia@etsii.upm.es (Á.G.-B.); japorro@etsii.upm.es (J.A.P.); marcos.diaz@upm.es (M.D.); jlocana@etsii.upm.es (J.L.O.)
\* Correspondence: ignacio.angulo@upm.es; Tel.: +34-910676700

**Abstract:** In recent years, an increasing interest in designing magnesium biomedical implants has been presented due to its biocompatibility, and great effort has been employed in characterizing it experimentally. However, its complex anisotropic behaviour, which is observed in rolled alloys, leads to a lack of reliable numerical simulation results concerning residual stress predictions. In this paper, a new model is proposed to focus on anisotropic material hardening behaviour in Mg base (in particular AZ31B as a representative alloy) materials, in which the particular stress cycle involved in Laser Shock Processing (LSP) treatments is considered. Numerical predictions in high extended coverage areas obtained by means of the implemented model are presented, showing that the realistic material's complex anisotropic behaviour can be appropriately computed and—much more importantly—it shows a particular non-conventional behaviour regarding extended areas processing strategies.

**Keywords:** laser shock processing; anisotropy; residual stress; FEM analysis; Mg AZ31B alloy

---

## 1. Introduction

Since 1930, an increasing interest in magnesium alloys has been documented in the aerospace industry [1,2] as a light-weight material. It is attractive for the biomedical industry in implant design, since it is degradable, with outstanding mechanical properties in comparison with polymers [3–5]. This has led to a great number of studies regarding mechanical properties enhancement [6–8], corrosion resistance [5,7–11] and microstructure modifications [12–14], among others. Single Mg crystal responses to shock loading in various conditions and orientations is also documented, including the study of the spall fracture [15], confirming the main mechanisms of hexagonal-closed-packed (hcp) crystals subject to uniaxial shock compression. In addition, the shock loading response of a single Mg crystal is compared with the one observed in a Mg alloy. The spall fracture results, from reference [15], were complemented in a recent publication with additional experiments at very high strain rates (up to $10^{-7} \cdot s^{-1}$) [16].

Magnesium alloys are presented in two different forms: The first ones are called casting alloys, which have defects such as inclusions or pores. The second ones are called wrought alloys, whose mechanical properties are enhanced with respect to casting alloys. This group includes rolled and extruded alloys. This paper is centered on the study of a rolled Mg AZ31B alloy, in which the grain is deformed along the rolling direction leading to a relevant asymmetry in its mechanical behaviour [17].

This anisotropic behaviour is motivated by the fact that different deformation modes are active depending on the loading path [18].

In general terms, a great effort has been employed in the experimental characterization of physical properties of magnesium alloys, but there is still a lack of simulation procedures regarding the accurate prediction of the residual stresses field. This may be motivated by the complex anisotropy and the atypical hardening behaviour of rolled Mg AZ31B alloy, in which great differences are observed in stress–strain curves in the normal direction (ND), rolling direction (RD) and transverse direction (TD). This anisotropic hardening has been studied by Tucker in Mg AZ31B alloy for different strain rates up to 4300 s$^{-1}$, in which the effect of the specimen temperature in results is also documented [19,20]. Jäger et al. [21] developed quasi static tensile behaviour experiments of hot rolled Mg AZ31 alloys up to 673 K. Biaxial tests were performed at room temperature to study the mechanical formability and elongation to failure [22]. Most of the researchers oriented these studies to understand the high-complexity physics involved in the different slip modes presented, twinning and detwinning in alternative paths, dislocations reorientation and the influence of grain size and texture in the mechanical properties in different loading orientations.

Regarding the developed mechanical constitutive models, several approaches to characterize stress–strain curves have been proposed in different situations [23,24]. Koh used Voce–Swift type isotropic hardening law to model the tensile stress–strain curves in RD and TD directions with temperatures ranging from 373 K to 573 K, with the purpose of evaluating the increased formability in hot rolled specimens [23]. Several authors used a visco-plastic self-consistent model to characterize Mg AZ31 alloys in a large variety of conditions [24]. However, a great number of constants need to be calibrated to fix these models. Thus, the identification of the particular stress cycles, temperatures and strain rates involved in LSP is necessary in order to develop a material model valid for residual stress determination.

In Laser Shock Processing (LSP), a high intensity pulsed laser beam irradiates the material's surface, forcing a sudden vaporization of the irradiated metallic target. The generated plasma expansion develops in a high intensity shockwave that propagates through the material leading to high strain rate deformations. The stresses in a material subject to LSP are characterized by alternative stress states in different orientations in the deviatoric plane. A hydrostatic pressure is simultaneously applied, which increases as the deviatoric stress does [25,26]. The hydrostatic pressure in metallic materials is usually neglected. However, several authors included an equation of state (Mie–Grüneisen) in their models [27,28], which correlates the applied pressure with the material's density. Nevertheless, the material's compressibility in the range of pressures involved in the considered alloy (from 0 to 5 GPa approximately) is properly defined by the Hooke's Law inherent to the elasto-plastic behaviour. Therefore, the hypothesis of neglecting the hydrostatic pressure is assumed and LSP will be considered as a process in which alternative stresses in different directions are applied.

Experimental evidence shows that anisotropic hardening behaviour occurs when alternative cycles are involved: In general, once a material is deformed by a compressive state, further lower amplitude tensile states result in plastic straining (Bauschinger effect). This is usually modelled by a kinematic hardening rule. Although the explicit consideration of the yield surface displacement within a saturated isotropic hardening rule is normally used to model aluminum alloys subject to LSP treatments in order to prevent the prediction of an undefined expansion of the yield surface [26], for the case of the considered alloy, reasonably good agreement is still possible for low-density treatments using a cyclicity-independent (i.e., conventional Johnson-Cook) hardening model.

In this paper, Hill's yield surface is used to properly model material hardening in both direct and reverse deformation paths. The reasonably good agreement between numerical residual stress predictions and experimental results for the highly anisotropic material considered (AZ31B alloy) confirms the appropriateness of the explicit consideration of anisotropic deformation behaviour in generic highly deformable materials.

## 2. Materials and Methods

### 2.1. Material Description

The material used in the present study is a 5 mm thickness plate of Mg AZ31B alloy (3 wt.% Al, 1 wt.% Zn, balance Mg) provided by Magnesium Elektron (Magnesium Elecktron Ltd., Manchester, UK). The Young modulus, *E*, is equal to 45 GPa. The Poisson ratio is 0.35. The anisotropic stress–strain curves along ND, RD and TD at quasi-static and dynamic conditions were adapted from Tucker's results [19], in which the stress–strain curves are represented for different strain rates. Considering that each of the stress–strain points in the original publication are not provided, a graph digitalization software was used to obtain each of the experimental results (Figure 1), which is necessary in order to develop and calibrate a material model.

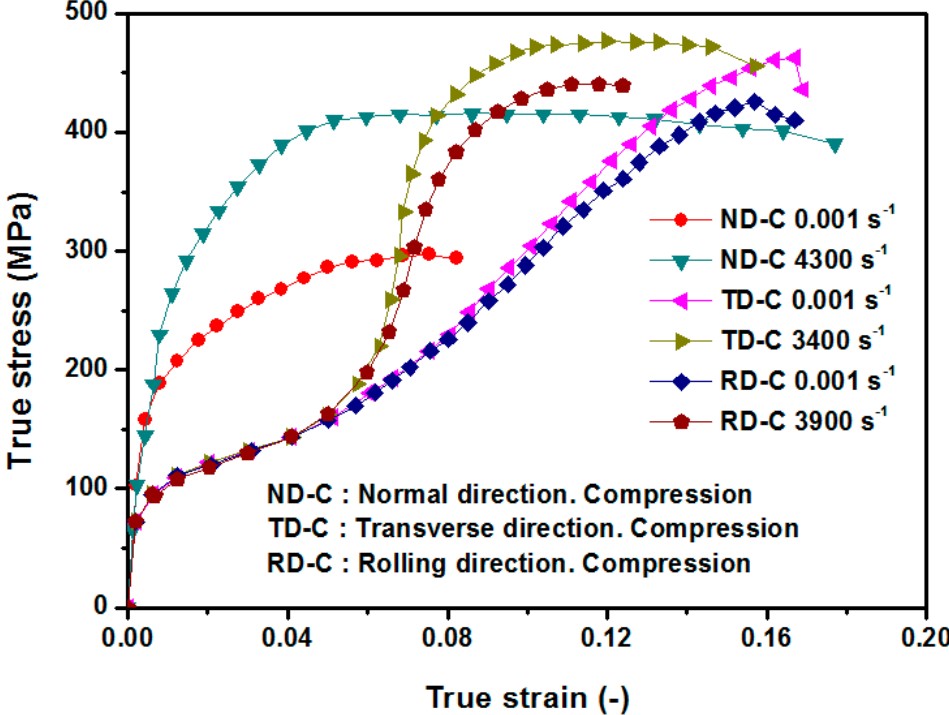

**Figure 1.** Stress–strain curves in rolled Mg AZ31B alloy at different strain rates adapted from [19].

### 2.2. Methodology

#### 2.2.1. Anisotropic Hardening Formulation based on Hill's Yield Surface to Model Alternative Loading Paths Presented in LSP

Classic $J_2$ plasticity theory assumes that the yield condition in stress space is governed by the second deviatoric stress invariant, $J_2$. This implies that the elastoplastic surface in stress space is considered as independent of the hydrostatic pressure and the third deviatoric stress invariant, $J_3$. The second invariant, $J_2$, is defined as a function of a generic stress state as follows:

$$J_2 = \frac{1}{6}\left[(\sigma_{22} - \sigma_{33})^2 + (\sigma_{11} - \sigma_{33})^2 + (\sigma_{11} - \sigma_{22})^2\right] + (\sigma_{23})^2 + (\sigma_{31})^2 + (\sigma_{12})^2 \tag{1}$$

The most widely implemented yield criterion based on $J_2$ theory used to model metallic materials is the von Mises criterion. It assumes that the material yields when the distortion energy per unit of volume reaches a limit. This limit is defined as the distortion energy achieved in a uniaxial tensile test, in which the yield condition is presented when $\sigma_1 = \sigma_y$ and $\sigma_2 = \sigma_3 = 0$, where $\sigma_y$ is the yield stress

of the material and $\sigma_1$, $\sigma_2$ and $\sigma_3$ are the three principal stresses. Thus, the yield surface results in a coaxial cylinder with the hydrostatic axis ($\sigma_1 = \sigma_2 = \sigma_3$) in the principal stresses space:

$$\left(\sigma_y\right)^2 = \frac{1}{2}\left[(\sigma_{22} - \sigma_{33})^2 + (\sigma_{11} - \sigma_{33})^2 + (\sigma_{11} - \sigma_{22})^2\right] + 3\left((\sigma_{23})^2 + (\sigma_{31})^2 + (\sigma_{12})^2\right) \tag{2}$$

$$\left(\sigma_y\right)^2 = \frac{1}{2}\left[(\sigma_2 - \sigma_3)^2 + (\sigma_1 - \sigma_3)^2 + (\sigma_1 - \sigma_2)^2\right] \tag{3}$$

In 1950, Hill extended the von Mises yield criterion to define anisotropic yield surfaces (Equation (4)), in which the parameters involved $F$, $G$, $H$, $L$, $M$ and $N$ can be defined as functions of the plastic strain. Thus, the material's elasto-plastic anisotropic behaviour can then be fully modelled. However, it is easier for researchers to define the yield stresses in three different directions, $\sigma_{yi}$ ($i = 1$ to (3)), expressed as a function of a generic yield stress, $\sigma_y$, where $R_{ii}$ are the first three potentials (Equation (5)). The other three potentials, $R_{ij}$ ($i, j = 1$ to 3, $i \neq j$, $i < j$), affect the material's behavior when shear stresses are presented. These potentials can be correlated with Hill's original parameters [29].

$$\left(\sigma_y\right)^2 = \frac{1}{2}\left[F(\sigma_{22} - \sigma_{33})^2 + G(\sigma_{11} - \sigma_{33})^2 + H(\sigma_{11} - \sigma_{22})^2\right] + 3\left(L(\sigma_{23})^2 + M(\sigma_{31})^2 + N(\sigma_{12})^2\right) \tag{4}$$

$$\sigma_{yi} = R_{ii}\sigma_y \tag{5}$$

In LSP, the plasma expansion develops in a high intensity shockwave that propagates through the material, which experiences a phenomenon called uniaxial strain [26], in which the material is subject to alternative loading paths (plastic loading and unloading). In the case of materials of highly anisotropic behaviour, as the considered case of AZ31B alloy, the proper modelling of these alternative loading paths requires an extended yield surface definition, in which the material's compressive behaviour along ND, RD and TD is properly modelled. Since the yield stress of the present alloy is relatively small, low-density treatments are expected to be the suitable ones to enhance its mechanical properties. Thus, the yield surface displacement, usually modelled by means of a kinematic hardening rule, will not be considered.

In the present paper, the plastic loading is modelled by means of the compressive stress–strain curve along the normal direction (ND), which is coincident with the direction of the applied pressure. Therefore, a biaxial compressive state in rolling and transverse directions (RD and TD) is considered during plastic unloading. Intuition suggests the use of the tensile stress–strain curve in the normal direction (ND) during plastic unloading. However, the effect of anisotropy in RD and TD would be neglected as a consequence, which is not realistic since different behaviour is expected along these directions.

### 2.2.2. Model Calibration Based on the Anisotropic Stress-Strain Curves of Mg AZ31B Alloy

The anisotropic stress–strain behaviour of Mg AZ31B alloy has been documented by many authors [30,31]. Several of them have focused their studies on warm temperatures (373 K to 573 K) since magnesium formability is relatively low at room temperature. Others have characterized the stress–strain curves in the ND subjected to uniaxial compressive states [20]. Concerning the cyclic behaviour, several authors studied the hysteresis loops in a wide variety of conditions: temperature, strain amplitude, directions and strain rates [17,30].

A great percentage of the stress–strain characterization works are centered on the RD and TD. The ND direction is harder to study due to its short length. In consequence, a lack of results is presented especially in the tensile direction. In spite of these difficulties, quasi static ND tensile stress–strain curves have been characterized in cyclic behaviour studies, in which the specimens were obtained from a 50 mm thickness rolled plate [18,30]. However, the dynamic behaviour at high strain rates is not characterized. In this section, an anisotropic model is calibrated based on the experimental dynamic characterization along ND, RD and TD.

The particularities of LSP processes need to be considered in order to select the most suitable experimental results to define the material model. Results need to include high strain rate dependence, while the effect of temperature and high compressive stress triaxialities could be ignored. In LSP, the explicit consideration of asymmetry in stress–strain curves may be considered for high-density treatments [26]. Due to the relatively small yield stress of AZ31B alloy, low-density treatments are expected to suitable and, consequently, the explicit consideration of asymmetry is neglected in the present study.

Regarding thermal effects at the material's surface, although the generated plasma usually reaches high temperatures, it is maintained during a very short period of time. Thus, the material is not expected to experience large temperature variations and room temperature is used for model calibration. For instance, the thermally affected depth for a steel alloy subject to LSP is estimated at 30 μm [32], while the plastically affected depth typically reaches more than 1 mm. As a result, thermal effects are usually neglected in LSP processes and the hypothesis of considering it as a purely mechanical process is generally adopted [32].

The experimental results that best fit with the above LSP characteristics definition are the ones presented by Tucker et al. [19]. These results include the quasi static and high strain rate behaviour in ND (only compression), RD (both tensile and compressive) and TD (both tensile and compressive) directions.

The plastic loading will be modelled by means of the compressive stress–strain curve in the ND. Therefore, compressive stress–strain curves in RD and TD characterize the biaxial stress state during plastic unloading. The yield stress of the present alloy is relatively small compared with most of the metallic materials subject to LPS treatments. As a consequence, low-density treatments are expected to be the suitable ones. This implies that the effect of the cyclic plasticity, modelled by means of a kinematic hardening rule, could be ignored. Thus, the yield surface in the present model will experience an anisotropic expansion in the stress space when the material is plastically deformed.

Regarding the stress–strain curve calibration along ND, a Voce type hardening law [33] has been used, including strain rate dependence (Equation (6)), where $\sigma_{ND}$ is the compressive yield stress in the ND, $\sigma_{ND0}$ is the first compressive yield stress, $Q$ is the yield stress saturated increment, $\dot{\varepsilon}_0$ is a reference strain rate, $b$ and $b_0$ are constants to be calibrated, $F(\dot{\varepsilon}_r)$ is a strain rate function that models the saturation equivalent plastic strain at different strain rates, and $\Delta\varepsilon_p$ is the equivalent plastic strain.

Considering that experimental results are obtained up to 4300 s$^{-1}$, the function $F(\dot{\varepsilon}_r)$ is set as constant for further strain rates, since there are no experimental data beyond 4300 s$^{-1}$. It is documented by some authors that extrapolating experimental results for higher strain rates usually leads to overestimations in the compressive residual stress predictions [34]. Table 1 presents the calibrated constants obtained for the best fit option. Figure 2 shows a comparison between the experimental results and the numerical predictions.

$$\sigma_{ND}\left(\Delta\varepsilon_p\right) = \sigma_{ND0} + Q\,F\left(\dot{\varepsilon}_r\right)\left(1 - e^{-(F(\dot{\varepsilon}_r)b + b_0)(\Delta\varepsilon_p)}\right) \tag{6}$$

$$F\left(\dot{\varepsilon}_r\right) = \begin{cases} 1 & \dot{\varepsilon}_r < \dot{\varepsilon}_0 \\[2mm] \dfrac{\dot{\varepsilon}_r}{\dot{\varepsilon}_1} + 1 & \dot{\varepsilon}_0 \le \dot{\varepsilon}_r \le \dot{\varepsilon}_1 \\[2mm] 2 & \dot{\varepsilon}_r > \dot{\varepsilon}_1 \end{cases} \tag{7}$$

**Table 1.** Calibrated constants for the analytic predictions of compressive stress–strain curves (ND).

| Parameter | Value |
| --- | --- |
| $\sigma_{ND0}$ (MPa) | 178 |
| $Q$ (MPa) | 125 |
| $b_0$ (–) | 19 |
| $b$ (–) | 18 |
| $\dot{\varepsilon}_0$ (s$^{-1}$) | 0.001 |
| $\dot{\varepsilon}_1$ (s$^{-1}$) | 4300 |

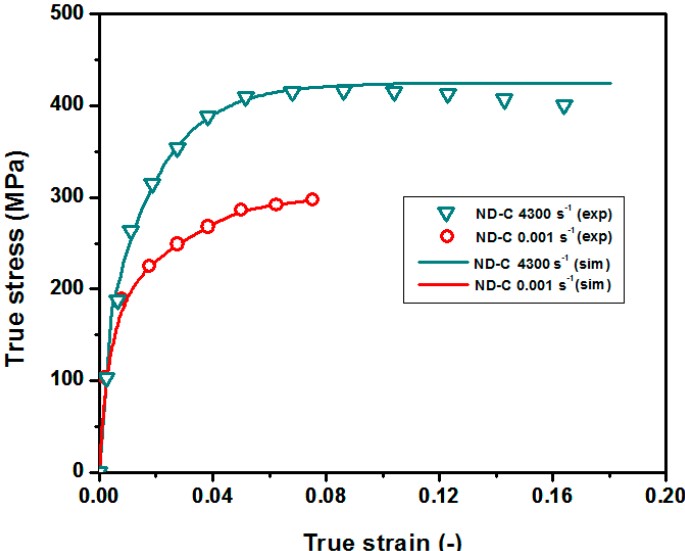

**Figure 2.** Calibrated model vs. experimental stress–strain curves along ND.

The material experiences softening at a certain plastic strain (approximately when the strain is equal to 0.07 at high strain rates), which is not possible to model with a purely Voce expression (monotonically increasing function). Koh used a Swift–Voce equation to predict softening [23]. However, in LSP, the equivalent plastic strain, $\Delta\varepsilon_p$, usually exceeds the material's ultimate tensile stress with no failure. This is due to the high compressive stress triaxialities, $\eta$, involved in LSP treatments, ensuring the complete protection of the material. In 2004, Bao and Wierbzicky [35] postulated a cut-off value for the compressive stress triaxiality beyond which failure will never occur ($\eta < -1/3$). This is the particular situation presented in LSP treatments. Therefore, a saturated hardening curve is considered to model the stress-strain scenarios beyond the ultimate tensile stress. Thus, Voce's equation itself predicts correctly this particular situation. It is important to point out that the isotropic models, which are widely accepted in LSP simulations (i.e., Johnson–Cook model) predict an unbounded expansion of the yield stress when high-density treatments are applied. This leads to overestimated compressive residual stress predictions, which are not observed experimentally. Thus, the use of a Voce equation (with saturated yield stress) may represent more accurate predictions than classic isotropic hardening models widely implemented in LSP simulations [26].

Regarding the constitutive equations to model the plastic unload, both RD and TD compressive stress–strain curves are considered. These curves present an atypical behaviour at an approximately equivalent plastic strain of 0.059, beyond which a new deformation mechanism is activated and yield stress increases abruptly. In order to model this behaviour, the superposition of two Voce equations will be used: The first one is characterized by a low amplitude. The second one, with higher amplitude, is activated once the equivalent plastic strain reaches the critical value. Equations (8)–(11) represent the stress–strain behaviour in RD and TD, both for quasi static and dynamic conditions. The dynamic behaviour from quasi static conditions up to the highest calibrated strain rates is obtained by linear

interpolation. The dynamic behaviour beyond the experimental data is set as constant. The calibrated parameters for the best fit option are shown in Table 2. Figures 3 and 4 show a comparison between the experimental results and the corresponding analytical predictions along RD and TD respectively.

$$\sigma_{RDe}\left(\Delta\varepsilon_p\right) = min\left(\sigma_{D0} + Q_D\left(1 - e^{-b_D(\Delta\varepsilon_p)}\right) + \partial\left(\Delta\varepsilon_p\right)\cdot Q_{RD2}\left(1 - e^{-b_{De}(\Delta\varepsilon_p)}\right), \sigma_{RDmax}\right) \tag{8}$$

$$\sigma_{TDe}\left(\Delta\varepsilon_p\right) = min\left(\sigma_{D0} + Q_D\left(1 - e^{-b_D(\Delta\varepsilon_p)}\right) + \partial\left(\Delta\varepsilon_p\right)\cdot Q_{TD2}\left(1 - e^{-b_{De}(\Delta\varepsilon_p)}\right), \sigma_{TDmax}\right) \tag{9}$$

$$\sigma_{RDd}\left(\Delta\varepsilon_p\right) = \sigma_{D0} + Q_D\left(1 - e^{-b_D(\Delta\varepsilon_p)}\right) + \partial\left(\Delta\varepsilon_p\right)\cdot Q_{RDd}\left(1 - e^{-b_{RDd}(\Delta\varepsilon_p)}\right) \tag{10}$$

$$\sigma_{TDd}\left(\Delta\varepsilon_p\right) = \sigma_{D0} + Q_D\left(1 - e^{-b_D(\Delta\varepsilon_p)}\right) + \partial\left(\Delta\varepsilon_p\right)\cdot Q_{TDd}\left(1 - e^{-b_{TDd}(\Delta\varepsilon_p)}\right) \tag{11}$$

$$\partial\left(\Delta\varepsilon_p\right) = \begin{cases} 0, & \Delta\varepsilon_p < \left(\Delta\varepsilon_p\right)_c \\ 1, & \Delta\varepsilon_p \geq \left(\Delta\varepsilon_p\right)_c \end{cases} \tag{12}$$

where $\sigma_{RDe}$ and $\sigma_{TDe}$ are the quasi static yield stresses along RD and TD respectively. $\sigma_{RDd}$ and $\sigma_{TDd}$ are the dynamic yield stresses. $\left(\Delta\varepsilon_p\right)_c$ is the critical equivalent plastic strain beyond which the high amplitude Voce function is activated.

**Table 2.** Calibrated constants for the analytic predictions of stress–strain curves.

| Parameter | Value |
|---|---|
| $\sigma_{D0}$ (MPa) | 60 |
| $\sigma_{RDmax}$ (MPa) | 426 |
| $\sigma_{TDmax}$ (MPa) | 463 |
| $Q_D$ (MPa) | 98 |
| $Q_{RD2}$ (MPa) | 720 |
| $Q_{TD2}$ (MPa) | 810 |
| $Q_{RDd}$ (MPa) | 280 |
| $Q_{TDd}$ (MPa) | 320 |
| $b_D$ (−) | 50 |
| $b_{De}$ (−) | 5 |
| $b_{RDd}$ (−) | 80 |
| $b_{TDd}$ (−) | 95 |
| $\left(\Delta\varepsilon_p\right)_c$ (−) | 0.059 |

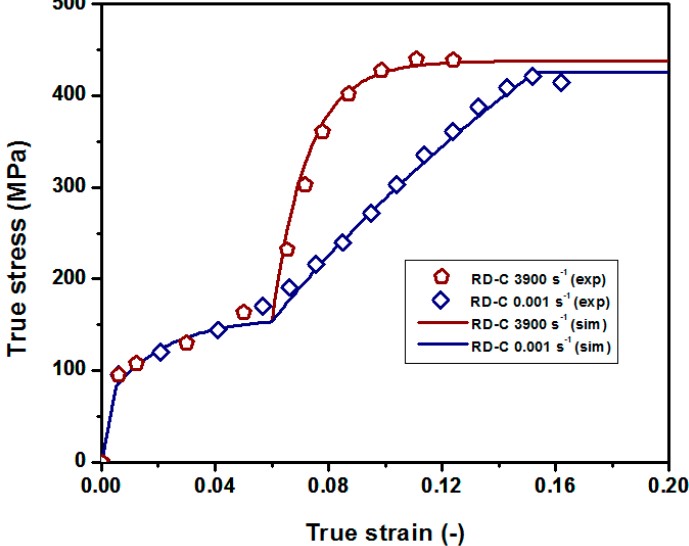

**Figure 3.** Calibrated model vs. experimental stress–strain curves along RD.

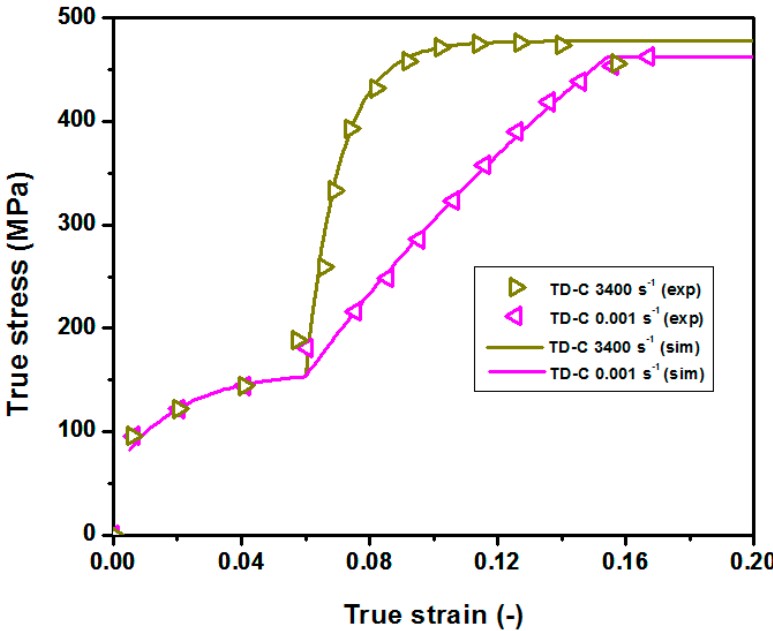

**Figure 4.** Calibrated model vs. experimental stress–strain curves along TD.

The calibrated stress–strain curves following the presented procedure are plotted together in Figure 5. The reference compressive yield stress, $\sigma_y$, is set equal to $\sigma_{D0}$. Thus, potentials, $R_{ij}$, are obtained dividing the corresponding yield stress in ND, TD and RD by the reference yield stress, $\sigma_{D0}$. The determination of Hill's coefficients is remarkably conditioned by the process, according to reference [36].

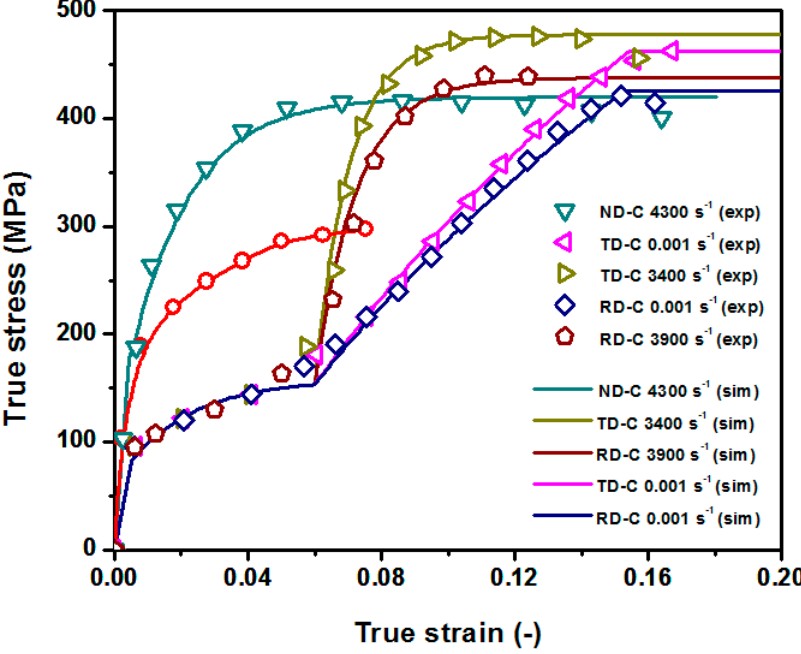

**Figure 5.** Calibrated model vs. adapted experimental stress–strain curves for ND, RD and TD.

2.2.3. Experimental Determination of the Spatial Pressure Pulse Profiles

In LSP, the high intensity laser beam applied to the surface forces a sudden vaporization generating a plasma, whose expansion develops in pressures about 5 GPa on the material's irradiated area. This

pressure generates a shockwave that propagates through the material, leading to plastic deformations and a final residual stress state. Thus, both the characterization of the plasma pressure and the simulation of the shockwave propagation through the material are key issues to model properly LSP treatments.

Regarding the plasma characterization, several authors adopt a simplified expression to estimate the maximum pressure as a function of laser intensity, in which the hypothesis of adiabatic expansion is considered [37]. However, in the present study, the plasma characterization is provided by a detailed calculation of plasma pressure and thermal flux (HELIOS code) as described in previous publications by the authors [38–40].

The laser device used in the present study is a Q-switched Nd:YAG (Spectra-Physics, Santa Clara, USA) operating at 10 Hz and providing 9 ns FWHM, 2 J per pulse with near Gaussian spatial profile. The spot diameter was set to 1.5 mm. This leads to a peak laser intensity, $I = 28.5$ GW/cm$^2$. The material's reflectivity was calculated with the aid of Wu and Shin model [41,42], which is set as an input parameter to the HELIOS code. The in-time pressure profile is represented in Figure 6.

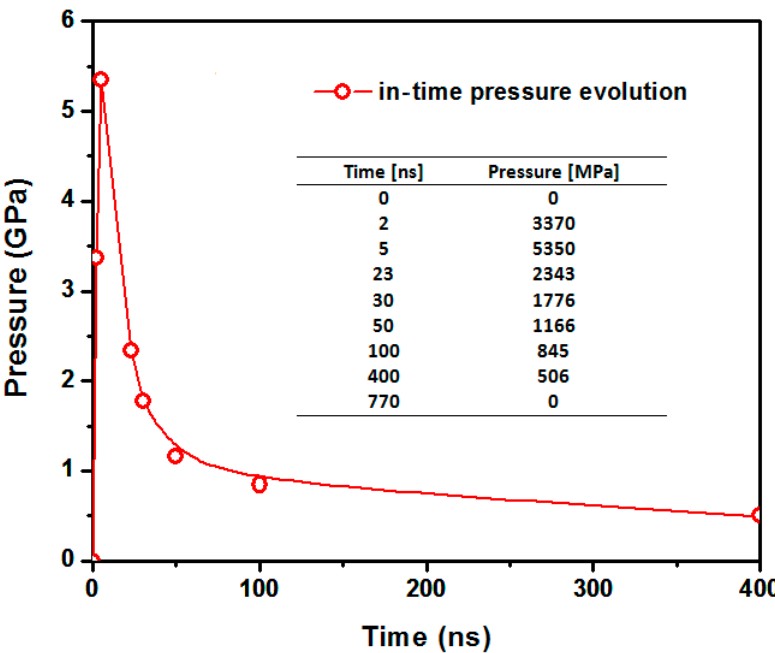

**Figure 6.** In-time pressure evolution calculated by means of HELIOS code.

The deformation profiles after the application of 1 to 5 concentric pulses in rolled Mg AZ31B alloy were obtained in the Laser Center UPM by means of a confocal microscope LEICA DCM 3D (Leica Microsystems GmbH, Wetzlar, Germany). In Figure 7, a characteristic map obtained for the application of a single pulse is presented.

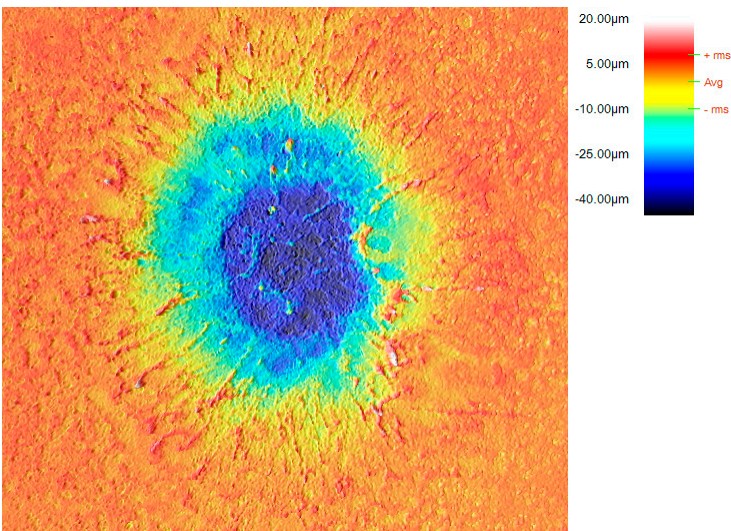

**Figure 7.** Confocal image of the deformed surface after the application of the first pulse.

The general form of the spatial distribution is defined by Equation (13), where $P(t)$ is the temporal pressure profile, $a_\phi$, $R_c$ and $H$ are constants to be calibrated, and $r$ represents the distance from a generic surface point to the center of the laser spot. In Figure 8, a comparison between the experimental softened profiles and its corresponding numerical predictions for the best fit option is plotted. Table 3 presents these calibrated constants.

$$P(r,t) = P(t)e^{-a_\phi\left(\frac{r}{R_c}\right)^{2H}} \tag{13}$$

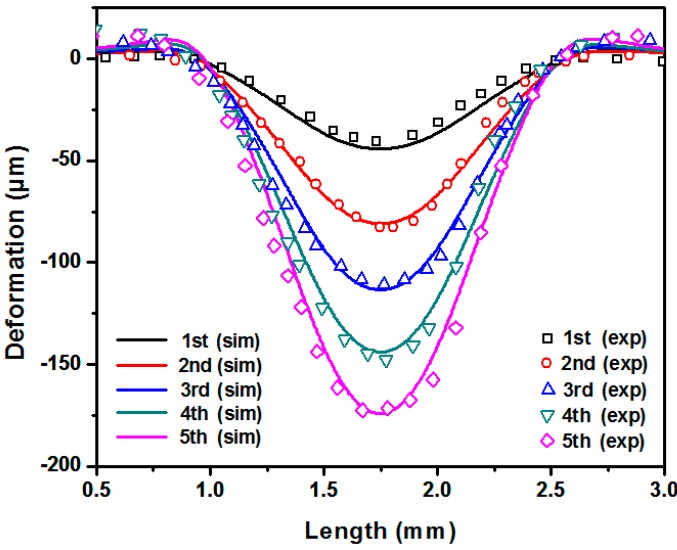

**Figure 8.** Experimental deformation profiles vs. numerical predictions.

**Table 3.** Calibrated constants for spatial pressure definition.

| Parameter | Value |
|---|---|
| $a_\phi$ (–) | 1.9 |
| $R_c$ (mm) | 0.8 |
| $H$ (–) | 1.3 |

## 3. Results

### 3.1. Experimental Characterization of the Residual Stresses for Different Input Parameters

The residual stresses have been obtained experimentally by means of the hole drilling method (ASTM E837-13 standard) [43], in which the material is drilled and the resulting deformation as the material is removed is measured by a precision strain gage up to 1 mm thickness (HBM K-RY61-1.5/120R-3 prewired). The specimens subject to LSP treatment were obtained from a 5 mm thickness rolled Mg AZ31B plate. Four different treatment strategies were implemented experimentally, covering a treated squared area of 900 mm² (Figure 9 shows a schematic representation):

(1) Setting a laser spot diameter of 1.5 mm, with a distance between successive pulses of 0.66 mm, developing in an Equivalent Overlapping Density, EOD [44], of 225 pp/cm², and an Equivalent Local Overlapping Factor, ELOF [44], of 4. The peening direction (PD) is set coincident with the transverse direction, TD.

(2) Setting a laser spot diameter of 1.5 mm, with a distance between successive pulses of 0.66 mm, developing in an Equivalent Overlapping Density, EOD, of 225 pp/cm², and an Equivalent Local Overlapping Factor, ELOF, of 4. The peening direction (PD) is set coincident with the rolling direction, RD.

(3) Setting a laser spot diameter of 1.5 mm, with a distance between successive pulses of 0.50 mm, developing in an Equivalent Overlapping Density, EOD, of 400 pp/cm², and an Equivalent Local Overlapping Factor, ELOF, of 7. The peening direction (PD) is set coincident with the transverse direction, TD.

(4) Setting a laser spot diameter of 1.5 mm, with a distance between successive pulses of 0.50 mm, developing in an Equivalent Overlapping Density, EOD of 400 pp/cm², and an Equivalent Local Overlapping Factor, ELOF, of 7. The peening direction (PD) is set coincident with the rolling direction, RD.

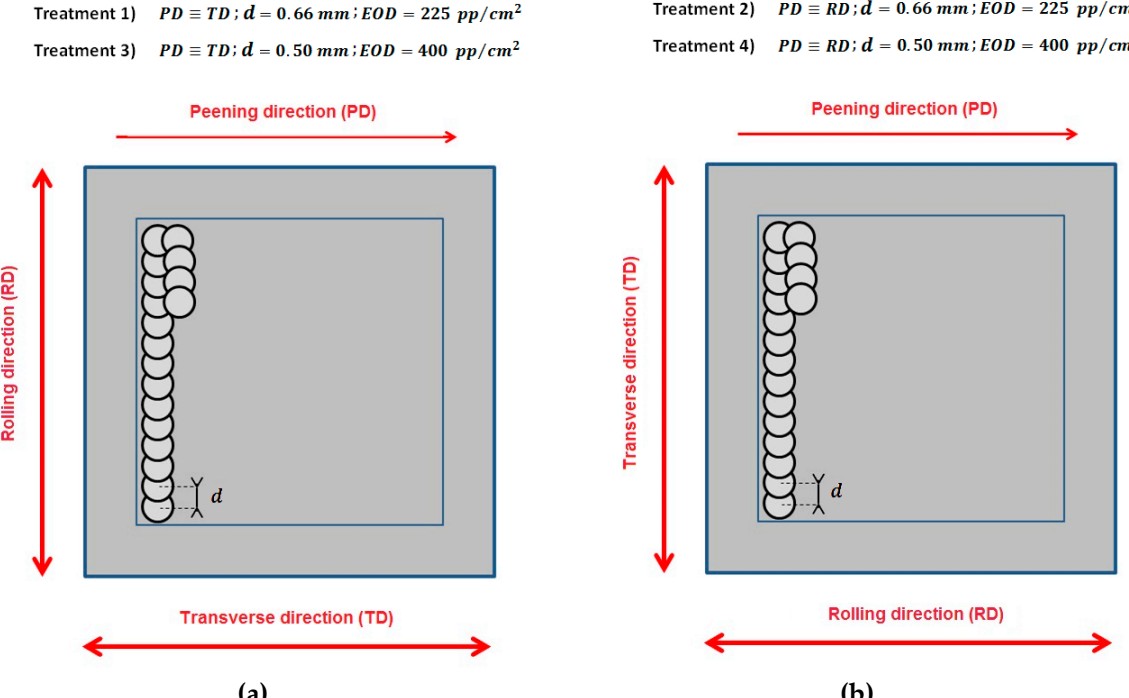

**Figure 9.** (**a**) Schematic representation of treatments (1) and (3) (PD = TD); (**b**) Schematic representation of treatments (2) and (4) (PD = RD).

The EOD corresponds to the number of applied pulses per unit of area, while the ELOF represents the average number of pulses at a given location. All the studied configurations can be considered as low-density treatments according to their respective EOD's and ELOF's. This is consistent with the proposed material model, in which the cyclic plasticity modelling is neglected. Higher density treatment-modelling would require further research in the proper calibration of stress–strain asymmetry.

The experimental results from the material's surface up to 1 mm for configurations (1), (2), (3) and (4) are represented in Figures 10 and 11. In these figures, the in-depth minimum, $S_{min}$, and maximum in plane, $S_{max}$ are represented with their corresponding uncertainties, calculated with the aid of Hdrill software. Several conclusions can be extracted from the experimental results:

(1) For all the experimental measurements developed, the minimum stress, $S_{min}$, is almost coincident with the stress along the PD, while the maximum in-plane stress, $S_{max}$, is similar to the stress along a perpendicular direction to the ND and PD This result is independent of the relative orientation between the PD and the RD.

(2) While significant differences are observed between treatment (1) and the others, great similarities are presented between configurations (2), (3) and (4). This suggests that the residual stresses tend to reach a saturation value for relatively low-density treatments, which is as expected considering the relatively low yield stress of the present alloy in comparison with most of metallic materials.

(3) Setting the peening direction (PD) coincident with the rolling direction (RD) leads to greater peak compressive residual stresses for low-density treatments (Figure 10).

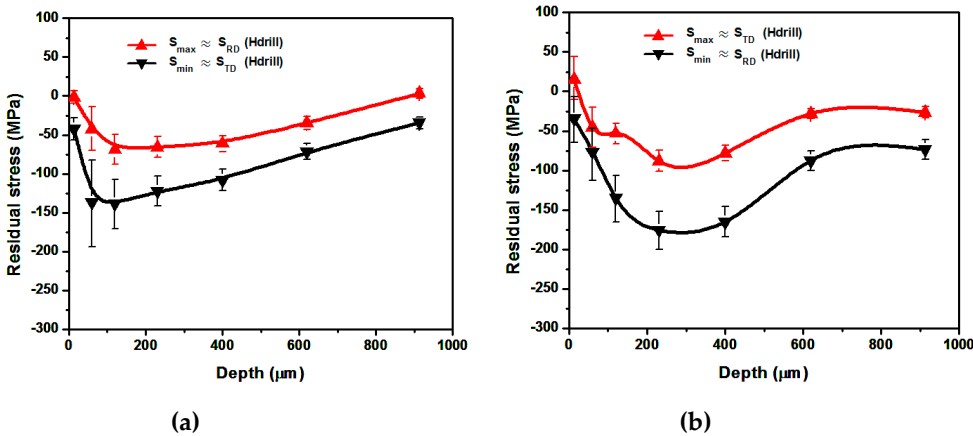

**(a)**　　　　　　　　　　　　　　　　　　　　**(b)**

**Figure 10.** (**a**) Experimental residual stresses for EOD = 225 pp/cm$^2$, PD = TD. (**b**) Experimental residual stresses for EOD = 225 pp/cm$^2$, PD = RD.

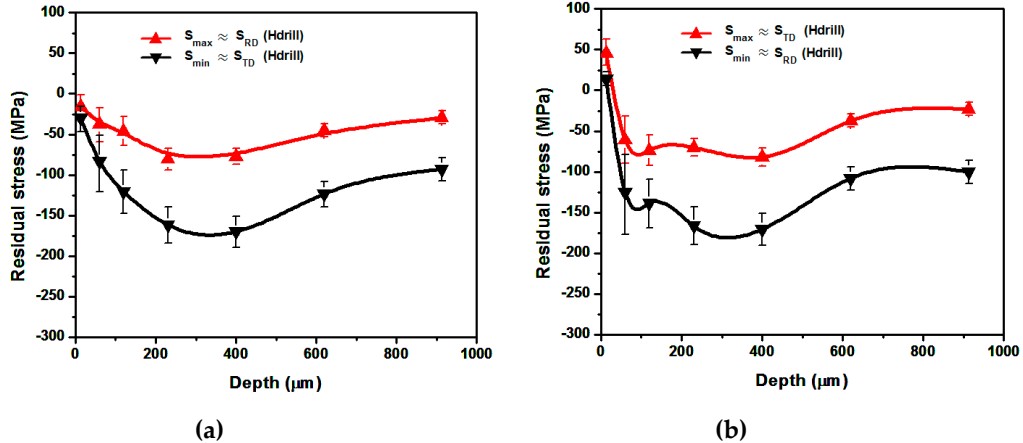

**(a)**　　　　　　　　　　　　　　　　　　　　**(b)**

**Figure 11.** (**a**) Experimental residual stresses for EOD = 400 pp/cm$^2$, PD = TD; (**b**) Experimental residual stresses for EOD = 400 pp/cm$^2$, PD = RD.

### 3.2. Realistic Modelling Results for Extended Surface High-Coverage LSP Treatments

The advances in computational resources in the last decade have motivated an increase in the number of publications regarding numerical predictions of extended LSP treatments in steel, aluminum and titanium alloys, with different loading strategies. However, there is still a lack of results concerning magnesium alloys which may be caused by the significant anisotropy presented. In this section, the corresponding numerical predictions of four particular treatments are obtained by means of two different models: (1) Considering the anisotropic model presented in Section 2.2, in which the stress–strain behaviour along ND, RD and TD directions is fully characterized. (2) Considering a purely isotropic model, in which both the plastic loading and unloading are modelled by the compressive stress–strain curve in ND direction. The purpose of comparing these two models is to show the impact in numerical predictions of the explicit consideration of anisotropy, since conventional LSP models consider a unique stress–strain curve. Results confirm that representative differences are predicted between the anisotropic model and the isotropic one. Better agreement with experimental results is obtained by means of the anisotropic model.

In realistic high-coverage extended treatments, such as the ones presented in this study, the peening direction motivates anisotropy in the residual stress predictions even in isotropic materials. Concretely, higher compressive residual stresses are usually predicted numerically and obtained experimentally along the PD [45]. This effect is also presented in the considered anisotropic alloy, in which results show that the minimum principal stress direction is coincident with the PD. The relative orientation between the PD and RD motivates variations in $S_{max}$ and $S_{min}$ which cannot be predicted by purely isotropic hardening models.

The numerical predictions have been extracted and averaged from a representative squared area of 2 mm$^2$ up to 1 mm depth, which represents a similar volume than the one removed by means of the hole drilling method. Figures 12–15 show a comparison between analytical predictions and experimental results for strategies (1), (2), (3) and (4) respectively. Table 4 shows a comparison between experimental results and the analytical predictions obtained by means of the anisotropic model. $(d_{max})_i$ is the depth at which the maximum peak compressive residual stress is presented. $min(S_{min})_i$ represents the peak compressive residual stress. $i = isot$ and $i = anisot$ corresponds to the analytical predictions of isotropic and anisotropic models respectively, and $i = exp$ to the experimental results.

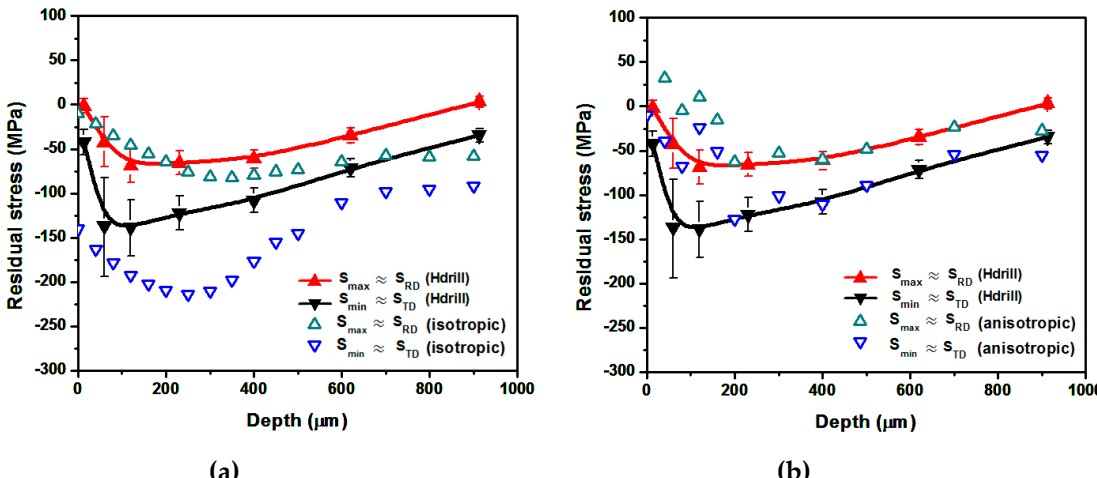

**Figure 12.** Experimental results vs. numerical predictions for strategy (1). (**a**) Analytical isotropic model predictions. (**b**) Anisotropic model predictions.

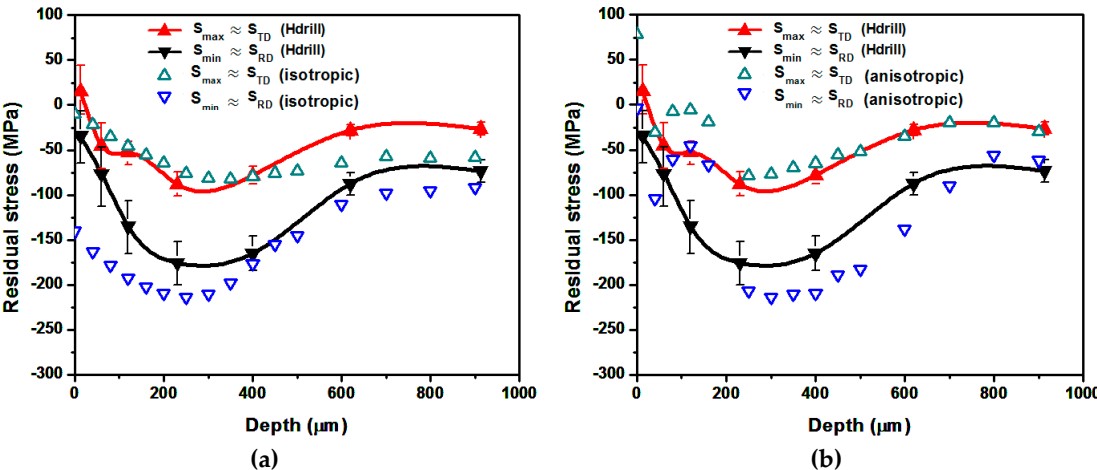

**Figure 13.** Experimental results vs. numerical predictions for strategy (2). (**a**) Analytical isotropic model predictions. (**b**) Anisotropic model predictions.

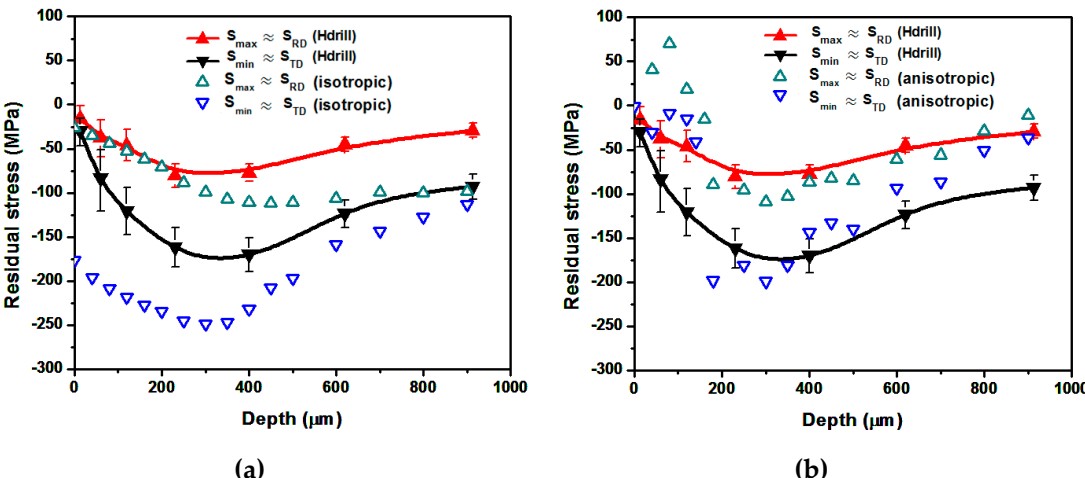

**Figure 14.** Experimental results vs. numerical predictions for strategy (3). (**a**) Analytical isotropic model predictions. (**b**) Anisotropic model predictions.

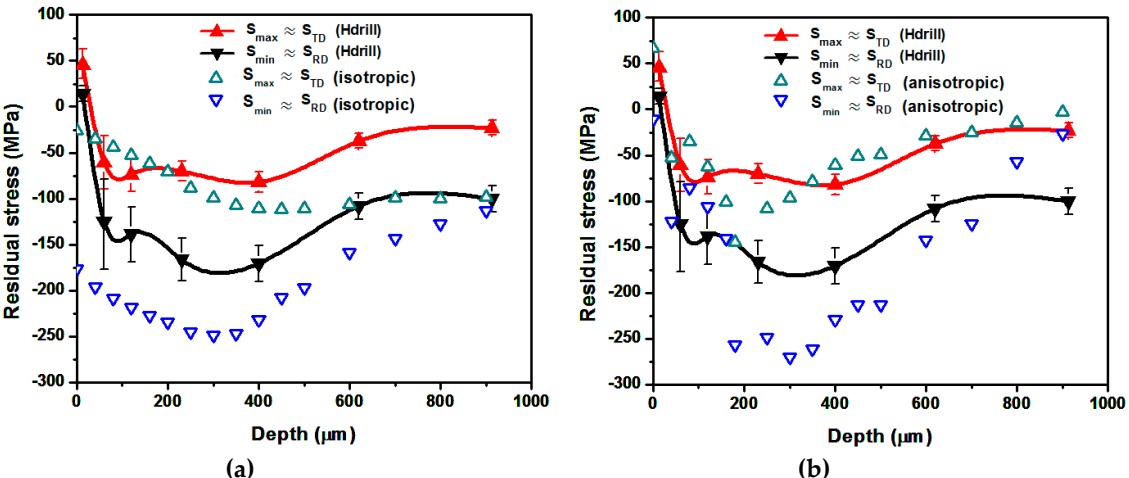

**Figure 15.** Experimental results vs. numerical predictions for strategy (4). (**a**) Analytical isotropic model predictions; (**b**) Anisotropic model predictions.

**Table 4.** Experimental results vs. analytical predictions for treatments (1), (2), (3) and (4).

| Feature | (1) | (2) | (3) | (4) |
|---|---|---|---|---|
| $EOD$ (pp/cm$^2$) | 225 | 225 | 400 | 400 |
| Peening direction (–) | TD | RD | TD | RD |
| $(d_{max})_{isot}$ (µm) | 250 | 250 | 299 | 299 |
| $(d_{max})_{anisot}$ (µm) | 197 | 297 | 297 | 300 |
| $(d_{max})_{exp}$ (µm) | 100 | 300 | 334 | 315 |
| $min(S_{min})_{isot}$ (MPa) | −215 | −215 | −249 | −249 |
| $min(S_{min})_{anisot}$ (MPa) | −130 | −213 | −199 | −268 |
| $min(S_{min})_{exp}$ (MPa) | −136 | −178 | −174 | −181 |

Regarding the principal stress directions, the PD sets the minimum stress curve, $S_{min}$. The maximum in plane stress, $S_{max}$, is perpendicular to the ND and the PD. This is predicted correctly by both models. However, the effect of the relative orientation between the PD and the RD in the stress curves, which is significant between the low-density treatments (strategies (1) and (2)), can only be simulated by the anisotropic model.

Concerning the comparison between experimental results and the analytical predictions, the anisotropic model provides a reasonably accurate calculation the effect of the relative orientation between the PD and RD for strategies (1) and (2). A comparison between Figures 12 and 13 shows that setting the PD coincident to the RD leads to greater peak compressive residual stresses. In addition, the depth at which this peak value is achieved also increases.

The anisotropic model provides, in general, more accurate predictions than the isotropic one. In particular, the isotropic model predicts high compressive residual stresses at material's surface, which is not consistent with experimental results. In treatments where the EOD is set equal to 400 pp/cm$^2$ (strategies (3) and (4)), experimental evidence shows that the residual stresses tend to saturate. Reasonably good agreement is obtained by means of the anisotropic model for strategy (3) while an overestimation of the compressive residual stresses is predicted for configuration (4). This is not surprising since the model is conceived for low-density treatments, which are the most suitable ones for the present alloy.

## 4. Discussion

This paper provides a complete methodology to obtain residual stress predictions in anisotropic materials subject to low-density LSP treatments. In summary: Firstly, the material's stress–strain curves in ND, RD and TD directions were properly modelled both at low and high strain rates. Then, the spatial-temporal pressure pulse distribution was calibrated with the aid of experimentally determined deformation profiles. Finally, the average in-depth residual stresses were obtained for four different treatment strategies, showing the effect of the treatment density and the relative orientation between the PD and RD in results. A reasonably good agreement is presented between the experimental results and the analytical predictions obtained by means of the proposed anisotropic model.

Experimental results suggest that the PD sets the minimum principal stress direction, which implies that the anisotropy in residual stress distribution is mainly motivated by the treatment strategy itself. In fact, the minimum stress, $S_{min}$, is obtained along the PD for the four treatment strategies experimented. This is properly simulated by both the isotropic and the anisotropic models. Nevertheless, the relative orientation between the PD and the RD motivates differences in the minimum stress curve, $S_{min}$, and the maximum stress, $S_{max}$, in low-density treatments. This effect cannot be predicted by the isotropic model, since a unique stress–strain curve is defined.

Regarding the comparison between the numerical predictions and the experimental results, better agreement is obtained by means of the implemented anisotropic model. This is observed specially at material's surface, where the isotropic model predicts high compressive residual stresses that are not presented in experimental results. The residual stress distribution from 200 µm to 700 µm is predicted

with great accuracy for configurations (1), (2) and (3). The greatest differences between the analytical predictions of the anisotropic model and the experimental results are presented near to the material's surface, where numerical results tend to predict higher tensile residual stresses than the ones observed experimentally. This may be caused by the evaporation of the material's surface during the application of successive pulses, leading to removal of already deformed material. This effect is not considered in LSP simulations [25–27,32,34,45].

The presented procedure is particularized for Mg AZ31B alloy. However, it is applicable to any material of interest subject to relatively low-density LSP treatments. In addition, the effect of the explicit consideration of mechanical anisotropy may differ depending on the selected material, since it is motivated by variations in the stress–strain curves along different loading paths.

## 5. Conclusions

This paper provides a complete guideline to model anisotropic behaviour of metallic materials subject to LSP treatments. Experimental evidence, confirmed by the corresponding numerical predictions, leads to the identification of the material's stress–strain anisotropy as a key factor to appropriately compute residual stress predictions in low-density treatments. The main conclusions are as follows:

(1) The relevance of including anisotropic behaviour in material modelling in residual stress predictions has been demonstrated in the particular case of Mg AZ31B alloy. Conventional isotropic models are not able to predict accurately the in-depth residual stress distribution. In addition, incorrect principal stress directions are obtained by means of purely isotropic models.

(2) The experimental results show that the anisotropy in residual stresses after treatment are motivated mainly by the treatment strategy. The minimum stress, $S_{min}$, is similar to the stress along the PD, while the maximum in plane residual stress, $S_{max}$, is similar than the one presented along a perpendicular direction to the ND and the PD. Both the implemented isotropic and anisotropic models predict properly this effect. Nevertheless, better agreement between numerical predictions and experimental results is achieved by the proposed anisotropic model.

(3) $S_{min}$ and $S_{max}$ curves are influenced by the relative orientation between the PD and the RD. Setting the PD coincident to the RD leads to greater compressive residual stresses for low-density treatments (EOD = 225 pp/cm$^2$). This effect cannot be computed properly by isotropic models. Therefore, the anisotropic hardening model is required for this purpose.

Regarding further developments, several issues may be considered:

(1) Modelling the material loss during target irradiation. The application of successive pulses leads to the evaporation of already deformed surfaces, which may modify residual stresses especially at material's surface.

(2) Analyzing the effect of the anisotropic modelling in residual stress predictions with a large variety of different input parameters, including additional relative orientations between the PD and material's anisotropic directions.

**Author Contributions:** Conceptualization, Á.G.-B. and J.L.O.; Data curation, F.C.; Investigation, J.A.P. and M.D.; Methodology, I.A.; Project administration, J.L.O.; Software, I.A. and F.C.; Supervision, J.L.O.; Validation, J.A.P., Á.G.-B. and M.D.; Writing—original draft, I.A. and F.C.; Writing—review and editing, I.A. and F.C. All authors have read and agreed to the published version of the manuscript.

**Funding:** Work partly supported by MINECO (Spain; Projects MAT2012-37782 and MAT2015-63974-C4-2-R).

**Conflicts of Interest:** The authors declare no conflict of interest.

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
