# Peer review of "Integrated Numerical-Experimental Assessment of the Effect of the AZ31B Anisotropic Behaviour in Extended-Surface Treatments by Laser Shock Processing"

_metals, doi:10.3390/met10020195_

Round 1

Reviewer 1 Report

In this work, the authors provides a complete methodology to model anisotropic behavior of metallic materials subject to LSP treatments. I recommend that this paper be accepted after minor revision.

1. In Figure 2, please describe the reason why the experimental curves and the calibrated model are different around 0.16 true strain. The trend of experimental curves seems to be decreasing.

In addition, It is stated that “The dynamic behavior beyond the experimental data is set as constant.” in Page 6, line 203. In this area, the constant horizontal line of calibrated model is correct?

2. Equation 19: Please define “r”.

3. Table 4: I would recommend to the authors to add the data of analytical isotropic model predictions.

4. It is stated that “This effect is usually neglected in LSP simulations.” in Page 15, line 407. Please add references to explain this sentence

5. General comment: I would recommend to the authors to check the format carefully. For instance, “All figures and tables should be cited in the main text as Figure 1, Table 1, etc.”, ” (Reference) Author 1, A.B.; Author 2, C.D. Title of the article. Abbreviated Journal Name Year, Volume, page range.”. Those sentences are from metals-template.

Reviewer 2 Report

In this paper, the authors have reported the investigation of Mg-based materials treated by laser shock. The scientific work comparison the mathematical prediction results with experimental results obtained on the materials after laser treatment. The experimental procedures and calculation methods used in this work very good described. The manuscript could be accepted for publication.

Author Response

Dear reviewer,

We are grateful for your quick answer and for the time and effort you have employed reviewing this paper.

Kind regards for your positive valorations.

Reviewer 3 Report

The authors consider LSP loading on rolled alloy AZ31B. The authors use a combined experimental-computational methodology to evaluate the residual stresses that develop due to LSP. The authors use a fairly primitive computational model, but it may be sufficient for the high strain rate, inertia-driven problem being studied. Some justification is needed for this approach. Due to the large amount of laser processing and interest in magnesium, this work will be of interest to the broader community and should be accepted for publication if the authors address the following issues. Small issues • Line 57, LSP is introduced without saying the full name • Eqns 1-11 may help the reader, but could also just be cited. These are common and contained in Hill’s book. Maybe just include essential eqns that appear later. • Compare Hill coefficients with other work on rolled AZ31B: Basu, Shamik, et al. "Towards designing anisotropy for ductility enhancement: A theory-driven investigation in Mg-alloys." Acta Materialia 131 (2017): 349-362. • Literature on laser shock of magnesium that should be included/considered: o Kanel, G. I., et al. "Shock response of magnesium single crystals at normal and elevated temperatures." Journal of Applied Physics 116.14 (2014): 143504. o De Rességuier, T., et al. "Spall fracture and twinning in laser shock-loaded single-crystal magnesium." Journal of Applied Physics 121.16 (2017): 165104. • In section 3.2 the authors use anisotropy and asymmetry interchangeably. Anisotropy refers to different strengths along different directions, whereas asymmetry refers to the case where the material is loaded along the same direction but exhibits different strengths in tension and compression. Larger issues • Phenomenological models that use plastic strain as the internal state variable that represents the instantaneous structural state generally do not do a good job of capturing cyclic loading. In particular, there is a broad body of literature that discusses the cyclic behavior of deformation twinning in magnesium. Some consideration/justification must be given for using a model that by construction does a poor job of representing cyclic loading, particularly when the deformation mechanisms that give rise to plastic anisotropy possess different reversibilities. • The authors reach temperatures and pressures where the material is vaporized. The anisotropy of AZ31B is well known to depend on temperature. During forming, it is generally observed deformation twinning, and therefore pronounced anisotropy, are not active above 200C. However, at high rates twinning may become prevalent again, even at elevated temperatures. The authors neglect fitting the temperature dependence of the anisotropy. The fact that the authors get reasonable agreement may be due to the fact that the problem is largely inertially driven, but some explanation for temperature dependence of anisotropy must be presented.
